# Effects of Repeated Short-Term Heat Exposure on Life History Traits of Colorado Potato Beetle

**DOI:** 10.3390/insects13050455

**Published:** 2022-05-12

**Authors:** Jianghua Liao, Juan Liu, Chao Li

**Affiliations:** Key Laboratory of the Pest Monitoring and Safety Control on Crop and Forest in Universities of Xinjiang Uygur Autonomous Region, College of Agronomy, Xinjiang Agricultural University, Urumqi 830052, China; ljh111196@163.com (J.L.); lj961527860@163.com (J.L.)

**Keywords:** *Leptinotarsa decemlineata*, high temperature, demographic parameters, fecundity, survival

## Abstract

**Simple Summary:**

Colorado potato beetle (CPB), is a serious pest of Solanaceae in China. High temperature under climate change is one of the main factors that affect the growth and fecundity of insects. In this study, CPB eggs and adults were repeatedly heat-treated at 35 °C and 39 °C for 1, 3 and 5 d (4 h a day), and at optimal 27 °C, and its egg hatching and adult reproduction were observed to explore the impact of short-term heat exposure on the population growth of CPBs. Our research has found that short-term heat exposure is not conducive to the development and reproduction of CPBs. These studies can contribute to deeper understanding how short-term heat exposure change CPBs fitness.

**Abstract:**

The Colorado potato beetle *Leptinotarsa decemlineata* (Say) (Coleoptera: Chrysomelidae) is an internationally recognized destructive pest which has caused serious losses to the potato industry. To clarify the impact of repeated short-term heat exposure on CPB egg hatching and adult fecundity under climate change, CPB eggs and adults were treated with repeated short-term heat exposure in this study. We found that the hatching rate of CPB eggs, the total number of eggs laid per female, the oviposition period, the intrinsic rate of population increase (*r*_m_), finite rate of increase (*λ*), and the net reproductive rate (*R*_0_) of CPBs decreased with increasing temperature. The hatching rate and fecundity of CPBs were significantly lower than those of control (CK) after repeated short-term heat exposure. Our research has found that repeated short-term heat exposure is not conducive to the development and reproduction of CPBs.

## 1. Introduction

In the context of global warming [1], the effect of environmental change on biology of organisms has become a hot topic of discussion [2]. Insects belong to ectotherm, their life activities are more sensitive to environmental change comparing to endotherms [3]. Temperature, as one of the main environmental factors affecting insect behavior [4,5], can have a profound impact on growth and development [6], survival [7], fecundity and other life activities [8].

Colorado potato beetles (*Leptinotarsa decemlineata*) were recognized worldwide as a devastating quarantine pest of potatoes, both the adult and larvae were able to bite out small holes in the leaf and gnaw at the edges of the leaves [9,10], and along with the spread of *Rhizoctonia solani* and ring rots. CPB was first discovered in north America in 1811. By the 1875s, CPB infesting potato fields across most of the United States and southern Canada. By the 1950s, CPB spread to Europe and Central Asia, Eastern Europe, Russia and Kazakhstan. CPBs invaded to Xinjiang in 1993 [11]. With the spread, it had now distributed to the north of Tianshan, posing a serious threat to the safe production of local potatoes.

The National Meteorological Administration stipulates that when the temperature is higher than 35 °C, it is a high temperature, and high temperature weather for more than 3 consecutive days is called high temperature heat wave [12]. Based on historical meteorological data from 1961 to 2015 in Xinjiang, the summer temperature in Xinjiang is on the rise. The temperature in 84.8% of Xinjiang could reach high temperatures above 35 °C, and some areas even reached 40 °C, and the temperature in 34.3% of meteorological stations of Xinjiang could reach high temperature weather lasting more than 10 days. This high-temperature environment has an important impact on local crops [13,14], and also affects the occurrence of pests [15,16,17].

The optimum temperature for CPB development is 25 °C–28 °C. The super-cooling point for CPB is −6 °C–17 °C [10,18]. When the temperature exceeds the optimum development temperature of the CPB, the life activities of the CPB will be impaired [19,20]. Studies found that the hatching rate of CPB eggs treated at 47 °C for 6 h were zero, and the emergence rate of pupae was close to 0 when the pupae was at 39 °C. CPB adults cannot survive for 1 h at 47 °C, and the mortality rate reaches 100% at 50 °C for 20–25 min. The high temperature above 39 °C, which lasted for more than one month, restricted the distribution area of CPBs [21,22,23]. Previous studies on the effects of high temperature on CPBs were mostly carried out under continuous high temperature or extreme lethal temperature conditions. High temperature in a natural environment cannot be sustained, and high temperature only lasts for some time [8,24]. In this paper, by studying effects of repeated heat exposure on CPBs, we can more deeply understand the survival and fecundity of CPBs, and this study can provide reference for exploring the occurrence regularity and forecasting of CPBs under the current conditions of increasing extreme meteorological conditions.

## 2. Materials and Methods

### 2.1. Insects Rearing

In this study, adults CPBs were collected from a potato field in Xishan Farm (43.6132° N, 87.3831° E, altitude: 1357 m), a suburb of Urumqi, Xinjiang. Adults CPBs were placed in culture dishes lined with moisturized filter paper and fed daily with fresh potato leaves in an artificial climate chamber. The climate chamber model is RXM-168C-1 climate chamber, it is produced by Ningbo Jiangnan Instrument Factory. The climate chamber was set at 27 ± 1 °C, 70% ± 5 RH, and a 16:8 h (L:D) photoperiod. The potato leaves used for CPBs reared were from potted seedlings planted under greenhouses. The temperature and light of greenhouses are consistent with the external environment. The plants are watered every two weeks without applying any pesticides. Field-collected CPBs were raised indoors to 2 generations. The adults collected outdoors are the first generation. The eggs laid by the outdoor adults develop into the adults is the second generation adults. Collect the eggs laid by the second generation adults for the experiment [25].

### 2.2. Treatments and Data Collection

The short-term high-temperature treatments selected were 35 °C or 39 °C at one of the above treatment temperatures for 4 h, and repeatedly treated for 1, 3, and 5 days. The hatching rate of CPB eggs is 0 under the continuous high temperature of 35 °C [26], and the emergence rate of the pupae is close to 0 under the continuous high temperature of 39 °C [21]. Therefore, in this study, 35 °C and 39 °C were selected as the research temperatures. High temperature weather for more than 3 consecutive days is called high temperature heat wave; therefore, in this study, the upper and lower values were taken as the number of high temperature stress days in 3 days. There were significant differences in the relative expression of Ld-hsp70a in adult CPBs treated at low temperature (−10 °C) and high temperature (44 °C) for 4 h; therefore, 4 h was chosen as the treatment time [27].

For study of the effects of repeated short-term heat treatments on egg development duration, we used fresh eggs produced within 24 h. Each group of 30–40 eggs was transferred in a 9 cm plastic culture dish, and the potato leaves moisturized with a moisturizing cotton ball were added for rearing. The culture dishes containing the eggs were placed in an artificial climate chamber. The climate box temperature was set to 39 °C ± 1 °C or 35 °C ± 1 °C; 70 ± 5% RH, and a 16:8 h (L:D) photoperiod. After being exposed for 4 h, the eggs were transferred to a separate artificial climate chamber (27 °C, 70 ± 5% RH, and a 16:8 h (L:D) photoperiod), allowed to hatch, and subjected to repeated short-term high temperature exposure for 1, 3, 5 days. Each different temperature was considered an experimental group, and each experimental group was replicated five times. The development and survival of the eggs were inspected and recorded every 24 h until all eggs were either black colored or had hatched.

For study of the effects of repeated short-term heat treatments on adult production, we used CPB adults within 24 h, with each pair of females and males placed in a 9 cm plastic culture dish, and the potato leaves moisturized with a moisturizing cotton ball were added for rearing. The culture dishes containing the CPB adults were placed in an artificial climate chamber. The climate box temperature was set to 39 °C ± 1 °C or 35 °C ± 1 °C; 70 ± 5% RH, and a 16:8 h (L:D) photoperiod. After being exposed for 4 h, they were then transferred to a separate artificial climate chamber (27 °C; 70 ± 5% RH, and a 16:8 h (L:D) photoperiod) and allowed to oviposit. The pre-oviposition duration, number of eggs oviposited per day, and longevity of the adults was inspected and recorded every 24 h until the death of all adults. The eggs were collected and removed daily. Fives pairs of females and males were used for each experimental group, and each experimental group was replicated with three replicates.

Regardless of the repeated short-term heat treatments of eggs or adults, they all used the temperature of 27 °C as the control (CK), and the climate box was set to 27 °C ± 1 °C; 70 ± 5% RH, and a 16:8 h (L:D) photoperiod.

### 2.3. Data Analyses

The data for the CPBs were analyzed using SPSS 26.0. The general linear model was used to analyze the test of between-subject effects of different treatment days and treatment temperatures. The data on the growth, development and oviposition of CPBs on different treatments were tested for normality and variance homogeneity. LSD multiple mean comparison method was used to compare significant differences among different treatments that met the assumptions. Nonparametric tests were used for the data that did not meet the assumptions, Kruskal–Wallis test was used, pairwise comparison method was used to compare significant differences among different treatments, and the *p* value is the value after Holm–Bonferroni correction. Because the hatching rate data conforms to the binomial distribution, logistic generalized linear model was used to analyze the hatching rate data (binomial distribution variable: successful hatching = 1, unsuccessful hatching = 0), and pairwise comparison method was used to compare significant differences among different treatments. The bar graphs in the figures were drawn using SigmaPlot 12.0.

The reproduction tables were established using the reproduction data of CPBs adults, and different treatment temperatures and days all had 3 replicates. Life table parameters analysis was as follows: the age-specific survival rate (*l_x_*), age-specific fecundity (*m_x_*), net reproductive rate (*R_0_*), intrinsic rate of increase (*r_m_*), finite rate (*λ*), and mean generation time (*T*) are calculated as follows:

The net reproductive rate (*R_0_*) is calculated as follows:$$R_{0}=\sum l_{x}m_{x}$$

The intrinsic rate of increase (*r_m_*) is calculated as follows:$$r_{m}=\frac{lnR_{0}}{T}$$

The finite rate (*λ*) is calculated as follows:$$\lambda =e^{r_{m}}$$

The mean generation time (*T*) is calculated as follows:$$T=\frac{\sum xl_{x}m_{x}}{\sum l_{x}m_{x}}$$

Through the above demographic parameters, we analyzed the effects of repeated short-term heat exposure on the growth of CPB population [28].

## 3. Results

### 3.1. Effects of Repeated Short-Term Heat Exposure on the Hatching of Colorado Potato Beetles Eggs

Different treatment days and treatment temperatures had no significant effect on the egg development duration of CPBs (*p* > 0.05). There was no significant interaction effect between treatment days and treatment temperatures on the egg development duration and the hatching of CPBs (*p* > 0.05). Different treatment temperatures had significant effect on the hatching of CPBs (*p* < 0.05) (Table 1); therefore, pairwise comparison method was used to compare significant differences among different treatment temperatures.

Repeated short-term heat exposure has significant effects on the hatching rate of CPBs (*p* ˂ 0.05). The hatching rate of CPBs decreased gradually with increasing temperature. The hatching rate of CPBs after repeated short-term heat exposure was significantly lower than that of CK (91%), and CPBs had the lowest hatching rate when they were exposed to repeated short-term heat at 39 °C (Table 2). Short-term high temperature had no significant effect on egg development duration of CPBs (*p* > 0.05) (Figure 1).

### 3.2. Effects of Repeated Short-Term Heat Exposure on the Reproduction of Adults Colorado Potato Beetles

Different treatment days had no significant effect on reproduction of CPBs. The interaction effect between treatment temperatures and treatment days was not significant on reproduction of CPBs (*p* > 0.05). Different treatment temperatures had significant effect on reproduction of CPBs (*p* < 0.05) (Table 3). Therefore, LSD multiple mean comparison was used to analyze the reproduction data among different treatment temperatures.

Repeated short-term heat exposure had no significant effect on the pre-oviposition duration of adults CPBs under different treatment temperatures (*p* = 0.852). Repeated short-term heat exposure had significant effect on the oviposition period for adults CPBs under different treatment temperatures (*p* = 0.000). The oviposition period gradually decreased with increasing temperatures. Repeated short-term heat exposure had a significant effect on the total number of eggs laid per female (*p* = 0.000). The total number of eggs laid per female gradually decreased with increasing temperature. The oviposition periods and the total number of eggs laid per female of CPB was significantly lower than those of CK (the oviposition period were 14.01 d; the total number of eggs laid per female was 315.96) after repeated exposure at 35 °C, 39 °C (Table 4).

### 3.3. Effect of Repeated Short-Term Heat Exposure on the Demographic Parameters of Adults Colorado Potato Beetle

Different treatment days had no significant effect on demographic parameters of CPBs, and the interaction effect between treatment temperature and treatment days was not significant (*p* > 0.05). Different treatment temperatures had significant effect on demographic parameters of CPBs (*p* < 0.05) (Table 5). Therefore, intrinsic rate of increase, finite rate of increase, mean generation time were analyzed with nonparametric tests, and net reproductive rate was used LSD multiple mean comparison in a general linear model to compare significant differences among different treatment temperatures.

There was no significant difference in mean generation time of CPB adults under repeated short-term heat exposure (*p* > 0.05). Repeated short-term heat exposure had a significant effect on the intrinsic rate of population increase in CPB (*p* = 0.000). The intrinsic rate of increase decreased with increasing temperature. Repeated short-term heat exposure had a significant effect on the finite rate of increase in CPB (*p* = 0.000). The finite rate of increase decreased with increasing temperature. Repeated short-term heat exposure had a significant effect on the net reproductive rate of CPB (*p* = 0.000), the net reproductive rate decreased with increasing temperature (Table 6).

## 4. Discussion and Conclusions

Temperature affects the development and survival of insects [29,30]. This study found that the hatching rate of CPB eggs at 27 °C is 91%, which is consistent with previous studies [31]. With the increase in the experimental treatment temperature, the hatching rate of CPB eggs gradually decreased. The hatching rate of CPBs under repeated short-term heat exposure was significantly lower than that of CK. This shows that repeated short-term high temperature is not conducive to the hatching of CPB eggs, previous studies also have found that short-term high temperature is not conducive to the hatching of CPB eggs. CPB eggs are treated at 35 °C for 8 h and the mortality rate is 31%, and the mortality rate is 100% under the continuous high temperature of 35 °C [25,26]. It has been also found that the emergence rate of CPB larvae was gradually decreased with the increasing of temperature [23], and in the 35~41 °C temperature range, the CPB survival rate decreased slowly. Starting from 44 °C, CPB survival rate decreased quickly with increasing exposure time, and in the first instar, no feeding occurred at 42 °C [23], indicating that high temperature is not conducive to CPBs survival.

This experimental study also found that repeated short-term high temperature affects the fecundity of adults CPB. The oviposition period of the CPBs at 27 °C was 14.01 days, and the total number of eggs laid per female was 315.96, which is in accordance with the results of the previous research on the female of the CPB, which produced 300–3130 eggs [32]. In this experiment, with the increase in the treatment temperature, the oviposition period and the total number of eggs laid per female of the CPB gradually decreased, which is consistent with the previous study showing that the reproductive capacity of the cotton aphid decreased with the increase in the treatment temperature under short-term high temperature [7]. It shows that short-term high temperature has an adverse effect on the egg production of CPB adults. Demographic parameters can be used to predict population growth: the intrinsic rate of increase reflects the growth ability of the population and can be used to measure the trend of population growth and decline at that time or in the future [29,33]. In this experiment, the intrinsic rate of increase in CPB at 27 °C was 0.16, which was significantly higher than repeated exposure at 35 °C and 39 °C, indicating that repeated short-term high temperature was not conducive to the growth of CPB population. Previous studies have found that the regulation of heat shock proteins can protect the body, but beyond a certain limit, the protective effect is limited [34,35]. This experiment found that the high temperature is not conducive to the fecundity and population growth of the CPB. Previous also found that short-term high temperature exposure had conducive effects on the fecundity of adult beetles: the fecundity of *Ophraella communa* significantly decreased with the increase in the treatment temperature, brief exposure to high temperatures during adulthood significantly reduces the fecundity of *Athetis lepigone* [36,37].

This study found that at 27 °C, the average pre-oviposition period of CPBs was 10.9 d longer than previous studies of CPBs reared at 27 °C. One possible reason is that the host species used are inconsistent, resulting in different nutrition [38,39].

After exposure of CPB eggs to short-term heat exposure, the egg period was not significantly different from CK. After exposure of CPB adults to short-term heat exposure, the pre-oviposition duration was not significantly different from CK of CPBs. On the other hand, the egg period of CPB is inconsistent with previous studies found that the egg period of CPBs under long-term stress at 31 °C is significantly shorter than that under 27 °C treatment [31]. Similarly to our results, repeated high-temperature exposure of *Propylaea japonica* at 39 °C had no significant effect on the pre-oviposition period [40]. This may be related to the treatment time. Insects need to accumulate temperature to complete a stage of development (effective accumulated temperature) [41].

Based on the analysis of the temperature data of 19 meteorological stations in Northern Xinjiang, from July to September 2016 to 2021 (the CPB occurrence period), it was found that there was a high temperature of 35 °C and 39 °C or higher in northern Xinjiang. In the past five years, the occurrence of high temperature at all stations showed an increasing trend, and the frequency of 3 consecutive days (or above) also showed an increasing trend (Figure 2). This experiment studied CPB eggs hatching and adult reproduction through indoor repeated short-term high temperature treatment. It can be analyzed that repeated short-term high temperature is not conducive to CPB eggs hatching and adult fecundity, which can provide a reference for more accurate and effective control strategies of the CPB. If high temperature is encountered during CPB control, the application can be appropriately reduced, but there are actually temperature fluctuations in the field, and CPBs have taken measures to reduce food intake and burrow into the soil in order to avoid high temperatures [42,43].

Future research also needs to further study the biological characteristics of CPB under temperature fluctuation environment and outdoor field. Under the global climate change, in the quarantine of CPBs, if the method of high temperature treatment is adopted, it should be considered that the heat resistance of CPBs will increase after high temperature domestication. Under global warming, the distribution area of CPBs may expand to high altitude areas. Therefore, future control and monitoring of CPBs should take into account global climate change [11,44].

## Figures and Tables

**Figure 1 insects-13-00455-f001:**
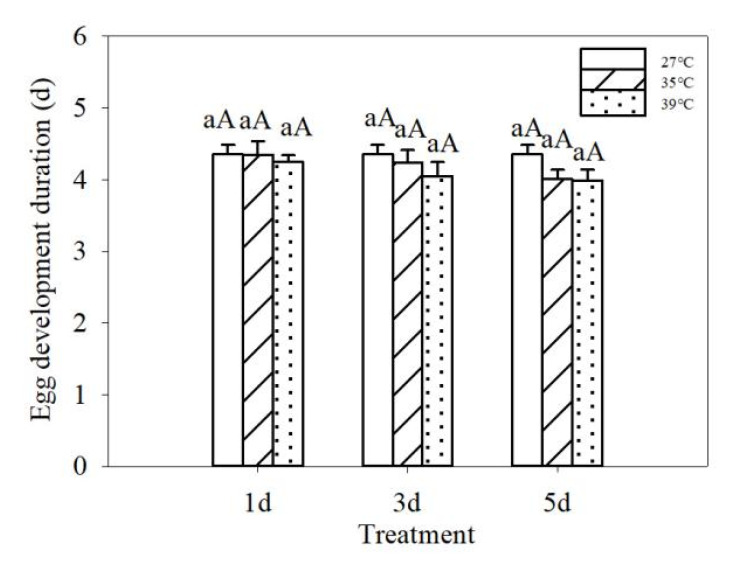
Egg development duration of *Leptinotarsa decemlineata* after the eggs were repeatedly exposed to different high temperature (35 °C and 39 °C) for 4 h during 1, 3 and 5 days. Data are expressed as mean ± SE. Data followed by the same letter in a histogram are not significantly different (*p* > 0.05).

**Figure 2 insects-13-00455-f002:**
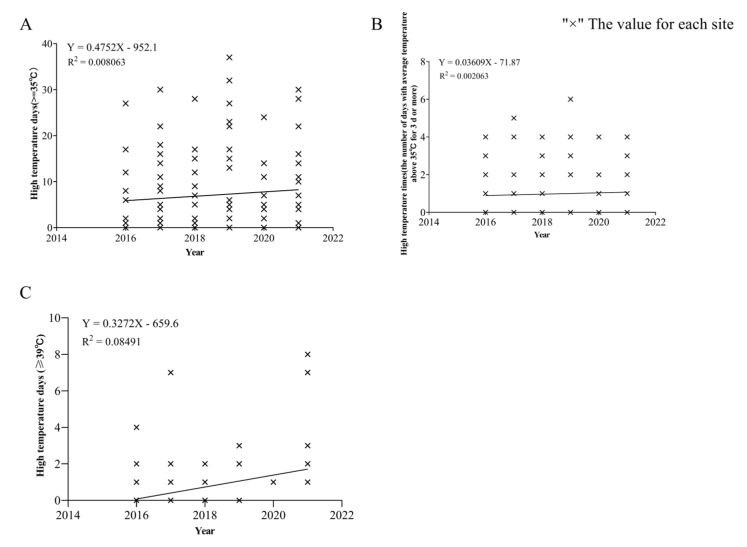
Analysis of high temperature at 19 stations in Northern Xinjiang. High temperature: Statistics of daily maximum temperature ≥ 35 °C at 19 stations in Northern Xinjiang from July to September 2016 to 2021. × is the value for each site. The line in the figure is a linear fit. (**A**) is the number of days when the temperature is higher than 35 °C at each station from 2016 to 2021; (**B**) is the number of times when the temperature is higher than 35 °C for 3 consecutive days or more at each station. (**C**) is the number of days when the temperature is higher than 39 °C at each station from 2016 to 2021.

**Table 1 insects-13-00455-t001:** Results of general linear model (GLM), generalized linear model for effects of treatment temperature, treatment days and their interaction effect on the egg development duration, the hatching rate of CPBs.

Independent Variable	Egg Development Duration			Hatching Rate		
	Degree Freedom	*F*	*p*	Degree Freedom	Wald χ^2^	*p*
Treatment temperature	2	1.278	0.287	2	192.883	0.000
Treatment days	2	0.714	0.494	2	3.282	0.194
Treatment temperature × Treatment days	4	0.223	0.924	4	1.652	0.799

**Table 2 insects-13-00455-t002:** Hatching rate of *Leptinotarsa decemlineata* after the adults were repeatedly exposed to different high temperature (27 °C, 35 °C and 39 °C). Data are expressed as mean ± SE. Data followed by the same letter in the same column are not significantly different (*p* > 0.05).

Treatment Temperatures	Hatching Rate (%)
27 °C (CK)	91.00 ± 1.20 a
35 °C	63.00 ± 1.90 b
39 °C	52.00 ± 1.80 c

**Table 3 insects-13-00455-t003:** Results of general linear model (GLM) for effects of treatment temperature, treatment days and their interaction on the reproduction of CPBs.

Independent Variable	Pre-Oviposition Duration			Fecundity			Oviposition Periods		
	Degree Freedom	*F*	*p*	Degree Freedom	*F*	*p*	Degree Freedom	*F*	*p*
Treatment temperature	2	0.162	0.852	2	171.889	0.000	2	59.377	0.000
Treatment days	2	0.790	0.471	2	1.847	0.190	2	3.497	0.055
Treatment temperature × Treatment days	4	0.473	0.755	4	0.964	0.454	4	1.045	0.415

**Table 4 insects-13-00455-t004:** Pre-oviposition duration, fecundity and oviposition periods of *Leptinotarsa decemlineata* after the adults were repeatedly exposed to different high temperatures (27 °C, 35 °C and 39 °C). Data are expressed as mean ± SE. Data followed by the same letter in the same column are not significantly different (*p* > 0.05).

Treatment Temperature	Pre-Oviposition Duration	Fecundity	Oviposition Periods
27 °C (CK)	10.92 ± 0.77 a	315.96 ± 4.44 a	14.01 ± 0.41 a
35 °C	10.80 ± 0.98 a	134.10 ± 16.04 b	8.62 ± 0.95 b
39 °C	11.69 ± 0.98 a	50.25 ± 4.08 c	3.90 ± 0.49 c

**Table 5 insects-13-00455-t005:** Results of general linear model (GLM) for effects of treatment temperatures, treatment days and their interaction on demographic parameters of CPBs.

Independent Variable	Intrinsic Rate of Increase *r*_m_			Finite Rate of Increase *λ*			Net Reproductive Rate *R*_0_			Mean Generation Time *T*		
	Degree Freedom	*F*	*p*	Degree Freedom	*F*	*p*	Degree Freedom	*F*	*p*	Degree Freedom	*F*	*p*
Treatment temperature	2	119.392	0.000	2	114.602	0.000	2	175.484	0.000	2	3.118	0.072
Treatment days	2	1.045	0.375	2	0.984	0.395	2	1.338	0.290	2	0.115	0.892
Treatment temperature × Treatment days	4	0.304	0.871	4	0.289	0.881	4	0.915	0.479	4	0.081	0.987

**Table 6 insects-13-00455-t006:** Demographic parameters of *Leptinotarsa decemlineata* after the adults were exposed to different high temperature (27 °C, 35 °C and 39 °C). Data are expressed as mean ± SE. Data followed by the same letter in the same column are not significantly different (*p* > 0.05).

Treatment Temperature	Intrinsic Rate of Increase *r*_m_	Finite Rate of Increase *λ*	Net Reproductive Rate *R*_0_	Mean Generation Time *T*
27 °C (CK)	0.1566 ± 0.0027 a	1.1695 ± 0.0032 a	160.1611 ± 2.6533 a	32.5043 ± 0.6518 a
35 °C	0.1332 ± 0.0019 b	1.1425 ± 0.0021 b	68.4073 ± 7.8663 b	31.3980 ± 0.2812 a
39 °C	0.1028 ± 0.0015 c	1.1083 ± 0.0017 c	23.5488 ± 1.3147 c	30.6224 ± 0.1611 a

## Data Availability

Data are available upon request from the authors.

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
