# Peer review of "Effects of Repeated Short-Term Heat Exposure on Life History Traits of Colorado Potato Beetle"

_insects, 2022, doi:10.3390/insects13050455_

Round 1

Reviewer 1 Report

Point 1: Considering the authors' previous paper "Duration of low temperature exposure affects egg hatching of the Colorado potato beetle and emergence of overwintering adults" published in Insects in 2021, it seems like an artificial split of the data to increase the number of publications. My opinion is that these data are more suitable for a short communication. Although if those data were combined, it would be complete scientific article.

Point 2:The methodology for describing the calculation of parameters analyzing the impact of short-term high temperature on the growth of CPB population refers to [26] Zhang, C.R.; B,an, X.F.; Shang, L.X.; Jun, J.; Liu, M.; Lan, L.S.; Mao, T.T.; Zhang, X.Y.; Zhi, J.R. Effects of Sitotroga cerealella on  the growth, development and reproduction of Orius strigicollis. J. Environ. Entomol. 2021, 43, 1016-1022. This article can not be found not by title, not by other imprint. In J. Environ. Entomol Volume 43 corresponds to 2014 and Volume 50 to 2021. However, this article is missing. Therefore, it is difficult to assess the applicability of some of the studied parameters.

Point 3: Modern references are very good, but for a problem that has been studied since the middle of the last century, I consider it necessary to mention older studies.

For example: Temperature-dependent Development and Feeding of Immature Colorado Potato Beetles Leptinotarsa decemlineata (Say) (Coleoptera: Chrysomelidae) PATRICK A. LOGAN, RICHARD A. CASAGRANDE, HEATHER H. FAUBERT, ANDFRANCIS A. DRUMMOND . 

Etc., and there are a lot of similar articles.

Point 4: I don't feel qualified to judge about the English language and style, but some sentences should be rewritten. 

For example: line 165 «The finite rate of  increase decreased with increasing temperature and treatment time;…..»

Author Response

Response to Comments

Point 1: Considering the authors' previous paper "Duration of low temperature exposure affects egg hatching of the Colorado potato beetle and emergence of overwintering adults" published in Insects in 2021, it seems like an artificial split of the data to increase the number of publications. My opinion is that these data are more suitable for a short communication. Although if those data were combined, it would be complete scientific article.

Response 1: Before the low temperature research, we had never considered doing a high temperature research. In the follow-up research, we thought that short-term high temperature is also a phenomenon that exists in reality, which has an impact on insects. Therefore, the high temperature test was carried out after the low temperature test was completed.

Point 2:The methodology for describing the calculation of parameters analyzing the impact of short-term high temperature on the growth of CPB population refers to [26] Zhang, C.R.; B,an, X.F.; Shang, L.X.; Jun, J.; Liu, M.; Lan, L.S.; Mao, T.T.; Zhang, X.Y.; Zhi, J.R. Effects of Sitotroga cerealella on the growth, development and reproduction of Orius strigicollis. J. Environ. Entomol. 2021, 43, 1016-1022. This article can not be found not by title, not by other imprint. In J. Environ. Entomol Volume 43 corresponds to 2014 and Volume 50 to 2021. However, this article is missing. Therefore, it is difficult to assess the applicability of some of the studied parameters.

Response2: I have replaced this document with another document. (Atashi, N., Shishehbor, P., Seraj, A.A., Rasekh, A., Hemmati, S.A., Riddick, E.W. Effects of Helicoverpa armigera egg age on development, reproduction, and life table parameters of Trichogramma euproctidis. Insects 2021, 12, 569.)

Point 3: Modern references are very good, but for a problem that has been studied since the middle of the last century, I consider it necessary to mention older studies.

For example: Temperature-dependent Development and Feeding of Immature Colorado Potato Beetles Leptinotarsa decemlineata (Say) (Coleoptera: Chrysomelidae)

Etc., and there are a lot of similar articles.

Response 3: I have added some similar articles.

Point 4:I don't feel qualified to judge about the English language and style, but some sentences should be rewritten.

For example: line 165 «The finite rate of increase decreased with increasing temperature and treatment time;…..»

Response 4: They have been modified.

Reviewer 2 Report

MAJOR REMARKS:

-Manuscript needs language editing by a native English speaker.

-Discussion should be more detailed.

TITLE:

-Title should be changed. It is not gramatically correct. It should not be in the form of question. It is not enough to say „short-term high temperature“ because it is repeated during several days. I suggest „Effects of repeated short-term heat exposure on life history traits of Colorado potato beetle“.

SIMPLE SUMMARY:

-Line 12: „and those cultured at 27℃” should be replaced with “and at optimal 27°C”.

-Lines 12-13: “and its egg hatching and adult reproduction were observed to explore the impact of short-term high temperature on the population growth of CPBs.” This should be separate sentence. Correct “short-term...” according to comments for the title.

-Lines 15-16: „this study...“ This should be separate sentence.

ABSTRACT:

-Line 17: Add „Say, Coleoptera: Chrysomelidae“ after „decemlineata“.

-Lines 18-19: Replace „continuous“ with „repeated“. The sentence is not complete.

-Lines 20-22: Same comments as for Simple summary.

-Lines 23-25: instead of two sentences I suggest: „We found that the hatching rate of CPB eggs significantly decreased with increasing temperature and the treatment time.” Make similar corrections for sentences in lines 25-27 and 27-30.

-Line 28: Insert space before (r) and (l). r should be rm.

-Lines 31-32: Delete this sentence.

KEY WORDS

-„demographic parameters“ instead of „life parameters“; lowercase letters for „Fecundity“

INTRODUCTION:

-Line 38: It should be „biology of organisms“.

-Line 39-40: It can be stressed that their life activities are more sensitive to environmental change comparing to endotherms.

-Line 43: Put space after „beetles“. Check for missing spaces in the whole text.

-Lines 50-53: The sentence is confused and should be rewritten.

-Line 55: Put „the“ instead of „he“.

-Line 59: Replace „inhibited“ with „impaired“.

-Lines 62-65: The statement is repeated twice.

-Line 67: Replace „normal“ with „natural“.

-Line 69: Replace „truly“ with „deeply“.

-Lines 68-71: The sentence is confused.

-What are the aims and hypotheses of the study? Which questions are posed?

MATERIAL AND METHODS:

-Line 75: Provide data on latitude, longitude and altitude for locality.

-Parts in parentheses in lines 78-79 and 81-83 should not be in parentheses. Write in normal sentences that ended with full stop.

-Lines 83-84: The sentence is not clear.

-Lines 86-88: It is not continuously. What do you mean with „respectively“? Explain why you used 35 and 39°C, why 4h, why up to 5 days. Can such values be recorded at examined locality?

-Lines 89-90: Avoid subtitles. I suggest „For study of the effects of repeated heat treatments on egg development duration we used fresh eggs produced within 24h“. Make similar correction for the begining of the next paragraph and in Data analysis (lines 112-115).

-Lines 93 and 104: It should be „or“ between 35 and 39°C. Why these parts are in parentheses?

-Line 96: Two parentheses should be deleted. It is not continuously and it is not only 5 days.

-Lines 96-97: I suggest „Each experimental group was replicated five times.“

-Line 99: Incubated? Do you mean hatched?

-Line 107: Respectively?

-Line 110: Replace „as one“ with „for each“.

-Line 112: Egg development duration? Demographic parameters? You did not mention oviposition traits.

-Lines 112-115: Do your data satisfy ANOVA asumptions? Which tests you used to check for normality and variance homogeneity?

-Lines 127-128: The sentence is not clear. It is repeated short-term exposure to high temperature.

RESULTS:

-Lines 133-135: Hatching rate is significantly lower in all treatment groups comparing to control, not only for 39°C during 5 days. It should be „duration of egg stage“ or „egg development duration“. Change that in title for Y axis.

-Lines 137-140: I suggest „Egg development duration (A) and egg hatching rate (B) of Leptinotarsa decemlineata after the eggs were repeatedly exposed to different high temperature (35 and 39°C) for 4h during 1, 3 and 5 days. Data are expressed as mean ± SE. Data followed by the same letter in a histogram are not significantly different (LSD test: p > 0.05).” Make similar corrections for caption to Figure 2.

-Line 145: Put full stop instead of comma before „With“.

-Y axis title for Figure 2B should be „Fecundity“ and for Figure 2C „Oviposition period (days)“. It seems to me that oviposition period differ significantly between 35 and 39°C after five days. However, both bars are marked with b. Please check this.

-Line 159: Demographic parameters?

-Put full stops between different sentences.

-Line 163: Insert „population“ before „increase“.

-Table 1: Demographic parameters. In the first column put comma between temperature and duration. In R0 column it seems to me that R0 significantly differ between 35 and 39°C after 5 days. However, values are marked with bc and c. Please check if it is correct.

DISCUSSION

-Line 182: Delete „4. Discuss“ at the begining.

-Line 184-185: „consistent with previous studies (29)“. Cite more studies.

-Line 188: It is not continuously.

-Lines 188-192: The sentence is too long and confused.

-Lines 192-195: There is not significant difference but it decreased gradually. I do not understand. Egg development duration was not affected by treatment. Maybe you can said that there was a trend of decrease.

-Line 196: Which development? Egg development? The sentence in lines 195-197 is confused.

-Line 198: Delete “egg”.

-Line 199: It should be “oviposition period”, not oviposition days.

-Line 200: It is not “each female”. It is “per female”.

-Line 201: 3130 eggs?

-Line 207: Demographic parameters.

-Lines 212-214: There are also other adaptive responses to high temperature, both physiological and behavioral.

-You should also mention studies on other species or CPB that are not consistent with your results and explain why those studies obtained different results.

-Mention studies that deal with CPB spreading in other countries.

-What are the implications of your results for CPB population dynamics? Although predictions depend on many factors not only temperature give some suggestions/expectations how CPB populations will respond to repeated hot days (migrations, selection of adapted individuals…).

Figure 3

-Line 234: It should be + instead of *.

-Line 237: Delete parenthesis.

REFERENCES:

-All journal names should be abbreviated.

-Line 261: Bradisia instead of bradisia.

-Line 266: Publisher is Academic Press.

-Species names should be in italic, e.g. in lines 284-285, 290-291, 301, 305, 323 etc.

-Line 289: It should be “…beetles’ hibernation”.

-Lines 300 and 304: Put “Insects” instead of “INSECTS”.

-Line 324: All article titles should be in sentence case.

Author Response

Response to Comments

Point 1: Title should be changed. It is not gramatically correct. It should not be in the form of question. It is not enough to say „short-term high temperature“ because it is repeated during several days. I suggest „Effects of repeated short-term heat exposure on life history traits of Colorado potato beetle“.

Response 1: The title has been modified.

Point 2:

SIMPLE SUMMARY:

-Line 12: „and those cultured at 27°C” should be replaced with “and at optimal 27°C”.

-Lines 12-13: “and its egg hatching and adult reproduction were observed to explore the impact of short-term high temperature on the population growth of CPBs.” This should be separate sentence. Correct “short-term...” according to comments for the title.

-Lines 15-16: „this study...“ This should be separate sentence.

Response 2: They have been modified.

Point 3: ABSTRACT:

-Line 17: Add „Say, Coleoptera: Chrysomelidae“ after „decemlineata“.

-Lines 18-19: Replace „continuous“ with „repeated“. The sentence is not complete.

-Lines 20-22: Same comments as for Simple summary.

-Lines 23-25: instead of two sentences I suggest: „We found that the hatching rate of CPB eggs significantly decreased with increasing temperature and the treatment time.” Make similar corrections for sentences in lines 25-27 and 27-30.

-Line 28: Insert space before (r) and (l). r should be rm.

-Lines 31-32: Delete this sentence.

Response 3: They have been modified.

Point 4:

 KEY WORDS

„demographic parameters“ instead of „life parameters“; lowercase letters for „Fecundity“

Response 4: They have been modified.

Point 5:

INTRODUCTION:

-Line 38: It should be „biology of organisms“.

-Line 39-40: It can be stressed that their life activities are more sensitive to environmental change comparing to endotherms.

-Line 43: Put space after „beetles“. Check for missing spaces in the whole text.

-Lines 50-53: The sentence is confused and should be rewritten.

-Line 55: Put „the“ instead of „he“.

-Line 59: Replace „inhibited“ with „impaired“.

-Lines 62-65: The statement is repeated twice.

-Line 67: Replace „normal“ with „natural“.

-Line 69: Replace „truly“ with „deeply“.

-Lines 68-71: The sentence is confused.

-What are the aims and hypotheses of the study? Which questions are posed?

Response 5: They have been modified.

The purpose of this article is to understand whether short-term high temperature has an impact on CPBs. If it does, high temperature factors should also be added to the prediction of the occurrence of CPBs to make predictions, and whether the control measures for CPBs at high temperature have changed and whether the use of chemicals can be reduced.

Point6:

MATERIAL AND METHODS:

-Line 75: Provide data on latitude, longitude and altitude for locality.

-Parts in parentheses in lines 78-79 and 81-83 should not be in parentheses. Write in normal sentences that ended with full stop.

-Lines 83-84: The sentence is not clear.

-Lines 86-88: It is not continuously. What do you mean with „respectively“? Explain why you used 35 and 39°C, why 4h, why up to 5 days. Can such values be recorded at examined locality?

-Lines 89-90: Avoid subtitles. I suggest „For study of the effects of repeated heat treatments on egg development duration we used fresh eggs produced within 24h“. Make similar correction for the begining of the next paragraph and in Data analysis (lines 112-115).

-Lines 93 and 104: It should be „or“ between 35 and 39°C. Why these parts are in parentheses?

-Line 96: Two parentheses should be deleted. It is not continuously and it is not only 5 days.

-Lines 96-97: I suggest „Each experimental group was replicated five times.“

-Line 99: Incubated? Do you mean hatched?

-Line 107: Respectively?

-Line 110: Replace „as one“ with „for each“.

-Line 112: Egg development duration? Demographic parameters? You did not mention oviposition traits.

-Lines 112-115: Do your data satisfy ANOVA asumptions? Which tests you used to check for normality and variance homogeneity?

-Lines 127-128: The sentence is not clear. It is repeated short-term exposure to high temperature.

Response 6:

(1)Lines 75, 78-79, 81-83, 83-84, 86-88, 89-90, 93, 104, 96, 96-97, 99, 107, 110, 112, 112-115, 127-128 have been modified.

(2)Lines 86-88: 35°С、39°С can be recorded in northern Xinjiang, and there may be 5 days of high temperature in northern Xinjiang. The hatching rate of CPB eggs is 0 under the continuous high temperature of 35°С, and the emergence rate of the CPB pupae is close to 0 under the continuous high temperature of 39°С. Therefore, in this study, 35°С and 39°С were selected as the research temperature. High temperature weather for more than 3 consecutive days is called high temperature heat wave, therefore, in this study, the upper and lower values were taken as the number of high temperature stress days in 3 days.

There were significant differences in the relative expression of Ld-hsp70a in adult CPBs treated at low temperature(-10°C) and high temperature(44°C) for 4 h.

The predecessors carried out short-term high temperature treatment on Calliptamus italicus( The dominant pest in the desert and semi-desert steppe pastoral areas of Xinjiang) and the treatment time was 4 h.( Xiang, M.; Fan, T.S.; Hu, H.X.; Yu,F.; Ji, R.; Wang, H. Effects of short-term exposure on the survival and fecundity of Calliptamus italicus(Orthopera: Acrididae). Chinese J. Appl. Entomol. 2017, 54, 426-433.)

(3)I checked the analytics data. The reproduction data of CPBs does not conform to normality and variance homogeneity, and the nonparametric test method has been used for data analysis. Levene test was used to check for normality and variance homogeneity.

Point 7:

RESULTS:

-Lines 133-135: Hatching rate is significantly lower in all treatment groups comparing to control, not only for 39°C during 5 days. It should be „duration of egg stage“ or „egg development duration“. Change that in title for Y axis.

-Lines 137-140: I suggest „Egg development duration (A) and egg hatching rate (B) of Leptinotarsa decemlineata after the eggs were repeatedly exposed to different high temperature (35 and 39°C) for 4h during 1, 3 and 5 days. Data are expressed as mean ± SE. Data followed by the same letter in a histogram are not significantly different (LSD test: p > 0.05).” Make similar corrections for caption to Figure 2.

-Line 145: Put full stop instead of comma before „With“.

-Y axis title for Figure 2B should be „Fecundity“ and for Figure 2C „Oviposition period (days)“. It seems to me that oviposition period differ significantly between 35 and 39°C after five days. However, both bars are marked with b. Please check this.

-Line 159: Demographic parameters?

-Put full stops between different sentences.

-Line 163: Insert „population“ before „increase“.

-Table 1: Demographic parameters. In the first column put comma between temperature and duration. In R0 column it seems to me that R0 significantly differ between 35 and 39°C after 5 days. However, values are marked with bc and c. Please check if it is correct.

Response 7:

Lines 133-135, 137-140, 145, 159, 16, Table 1 128 have been modified.

“It seems to me that oviposition period differ significantly between 35 and 39°C after five days. However, both bars are marked with b. Please check this.” 

Response : After I checked it carefully, it was correct

“In R0 column it seems to me that R0 significantly differ between 35 and 39°C after 5 days. However, values are marked with bc and c. Please check if it is correct.”

Response : After reanalysis of this part of the data using nonparametric tests, there is still no significant difference.

Point 8:

DISCUSSION

-Line 182: Delete „4. Discuss“ at the begining.

-Line 184-185: „consistent with previous studies (29)“. Cite more studies.

-Line 188: It is not continuously.

-Lines 188-192: The sentence is too long and confused.

-Lines 192-195: There is not significant difference but it decreased gradually. I do not understand. Egg development duration was not affected by treatment. Maybe you can said that there was a trend of decrease.

-Line 196: Which development? Egg development? The sentence in lines 195-197 is confused.

-Line 198: Delete “egg”.

-Line 199: It should be “oviposition period”, not oviposition days.

-Line 200: It is not “each female”. It is “per female”.

-Line 201: 3130 eggs?

-Line 207: Demographic parameters.

-Lines 212-214: There are also other adaptive responses to high temperature, both physiological and behavioral.

-You should also mention studies on other species or CPB that are not consistent with your results and explain why those studies obtained different results.

-Mention studies that deal with CPB spreading in other countries.

-What are the implications of your results for CPB population dynamics? Although predictions depend on many factors not only temperature give some suggestions/expectations how CPB populations will respond to repeated hot days (migrations, selection of adapted individuals…).

Figure 3

-Line 234: It should be + instead of *.

-Line 237: Delete parenthesis.

Response 8: 

They have been modified; 182, 188-192, 192-195, 196, 198, 199, 200, 207, 212-214, 234, 237 have been modified.

3130 eggs? It has been checked, it is correct. (Guo, W.C.; Tuerxun; Cheng, D.F.; Tan, W.Z.; Zhang, Z.K.; Li, G.Q.; Jiang, W.H.; Deng, J.Y.; Wu, J.H.; Deng, C.S.; Li, J.; Liu, X.X.; Lv, H.P. Main progress on biology &ecology of Colorado potato beetle and counter measures of its monitoring and controlling in China. Plant Protection 2014, 40, 1-11.)

Other comments have been changed as requested.

Point 9:

REFERENCES:

-All journal names should be abbreviated.

-Line 261: Bradisia instead of bradisia.

-Line 266: Publisher is Academic Press.

-Species names should be in italic, e.g. in lines 284-285, 290-291, 301, 305, 323 etc.

-Line 289: It should be “…beetles’ hibernation”.

-Lines 300 and 304: Put “Insects” instead of “INSECTS”.

-Line 324: All article titles should be in sentence case.

Response 9: They have been modified.

Round 2

Reviewer 1 Report

Point 1:

Line 227- 228 «It may be that  the high temperature destroys both physiological and behavioral of the CPB and causes  protein denaturation.»

I missed it in the first review. But the temperature 39C are not enough to denature proteins. Contrary, the authors should provide a references that confirms this.

Point 2: In response to the previous comment, the authors indicated that they have replaced the references, but this I did not see it:

«Point 2:The methodology for describing the calculation of parameters analyzing the impact of short-term high temperature on the growth of CPB population refers to [26] Zhang, C.R.; B,an, X.F.; Shang, L.X.; Jun, J.; Liu, M.; Lan, L.S.; Mao, T.T.; Zhang, X.Y.; Zhi, J.R. Effects of Sitotroga cerealella on the growth, development and reproduction of Orius strigicollis. J. Environ. Entomol. 2021, 43, 1016-1022. This article can not be found not by title, not by other imprint. In J. Environ. Entomol Volume 43 corresponds to 2014 and Volume 50 to 2021. However, this article is missing. Therefore, it is difficult to assess the applicability of some of the studied parameters.

Response2: I have replaced this document with another document. (Atashi, N., Shishehbor, P., Seraj, A.A., Rasekh, A., Hemmati, S.A., Riddick, E.W. Effects of Helicoverpa armigera egg age on development, reproduction, and life table parameters of Trichogramma euproctidis. Insects 2021, 12, 569.

Line 137 Through the above demographic parameters, analyze the effects of repeated short- term heat exposure on the growth of CPB population [29].

Line 340 29. Zhang, C.R.; Ban, F.X.; Shang, X.L.; Guo, J.; Liu, M.; Mao, T.T.; Zhang, X.Y.; Zhi, J.R. Effects of Sitotroga cerealella on the growth, development and reproduction of Orius strigicollis. J. Environ. Entomol. 2021, 43, 1016-1022.

Please check references and bibliography, delete non-existing articles : (29. Zhang, C.R.; Ban, F.X.; Shang, X.L.; Guo, J.; Liu, M.; Mao, T.T.; Zhang, X.Y.; Zhi, J.R. Effects of Sitotroga cerealella on the growth, development and reproduction of Orius strigicollis. J. Environ. Entomol. 2021, 43, 1016-1022. ).

The article «Atashi, N., Shishehbor, P., Seraj, A.A., Rasekh, A., Hemmati, S.A., Riddick, E.W. Effects of Helicoverpa armigera egg age on development, reproduction, and life table parameters of Trichogramma euproctidisInsects 2021, 12, 569.)»  is absent in the reference list.  Authors need to carefully check whole text and reference list.

Point 3:  At the first mention of «CK», it is necessary to explain the abbreviation.

Author Response

Point 1:

Line 227- 228 «It may be that the high temperature destroys both physiological and behavioral of the CPB and causes protein denaturation.», I missed it in the first review. But the temperature 39℃ are not enough to denature proteins. Contrary, the authors should provide a references that confirms this.

Response 1: They have been modified.

Point 2: In response to the previous comment, the authors indicated that they have replaced the references, but this I did not see it:

Please check references and bibliography, delete non-existing articles : (29. Zhang, C.R.; Ban, F.X.; Shang, X.L.; Guo, J.; Liu, M.; Mao, T.T.; Zhang, X.Y.; Zhi, J.R. Effects of Sitotroga cerealella on the growth, development and reproduction of Orius strigicollis. J. Environ. Entomol. 2021, 43, 1016-1022. ).

The article «Atashi, N., Shishehbor, P., Seraj, A.A., Rasekh, A., Hemmati, S.A., Riddick, E.W. Effects of Helicoverpa armigera egg age on development, reproduction, and life table parameters of Trichogramma euproctidis. Insects 2021, 12, 569.)» is absent in the reference list. Authors need to carefully check whole text and reference list.

Response 2: Sorry, they have been modified and we double checked the reference.

Point 3: At the first mention of «CK», it is necessary to explain the abbreviation.

Response 3: They have been modified.

Reviewer 2 Report

-There are still many missing spaces (e.g., lines 9, 72, 144, 156, 159, 162, 167, 172, 179, 248, 342, Y titles in Figure 1…

-Lines 15-16: Chnange sentence into “These studies can contribute to deeper understanding how short-term heat exposure change CPB fitness.“

-Line 22: Delete „by“, put „population“ before „increase“.

-Line 24: Put „…then those of control (CK) beetles“. Abbreviation CK appears for the first time in the text.

-Line 33: Put „comparing to endotherms“ at the end of sentence.

-Line 49: Do not start the sentence with the number.

- Lines 80-82: The sentence should not be in parenthesis. Put the separate sentence at the end of paragraph. Similarly the part in lines 85-93 should not be in parenthesis.

-Line 124: Correct „viposition“.

-Line 125: Which post hoc test did you use for non-parametric analyses? Is it SigmaPlot instead of SigmPlot?

-Line 127: Comma instead of full stop before „net“.

-Line 137: Replace „analyze“ with „we analyzed“.

-Line 141: Put p˂0.001.

-Line 145: Put „exposed to“ before „repeated“. Put „Short-term“ instead of „short-term“.

-Lines 15, 161, 162, 212, 215, : Delete „by“.

-Line 163: Sign for female is not needed.

-Lines 168, 191: LSD is not post hoc test for non-parametric analyses.

-Line 194: Delete „4. Discuss“ at the beginning of subtitle.

-Lines 204-205: Start with „It has been also found that the emergence rate….“

-Line 207: „Starting from 44℃ CPB survival rate…” should be new sentence.

-Line 208: Insert „exposure“ before „time“ and „in“ before „the first instar“. Replace „so“ with „pointing that“.

-Line 212: Replase „the same as“ with „in accordance with“.

-Line 213: Delete „per“.

-Line 216: Insert „showing“ before „that“.

-Line 218: „It shows that short-term….“ should be new sentence.

-Lines 227-229: Put „disturb other physiological and behavioral traits“ instead of „destroy…“. Cite some references at the end of the sentence.

-Line 231: Delete one „the“. Replace „raised“ with „reared“.

-Line 232: Part in parentheses can be deleted.

-Line 234: Change into: „After exposure of CPB eggs to short-term….“.

-Line 235: Change into: „After exposure of CPB adults to short-term….“.

-Lines 236-238: Change into: „On the other hand, the egg period of CPB…“. Cite reference.

-Lines 238-240: Change into: „Similar to our results repeated high-temperature exposure of…“.

-Lines 240-242: The sentence is not clear.

-Line 245: Put „temperatures“ instead of „temperature“. Put „and“ between 35 and 39. Put „higher“ instead of „above“.

-Line 255: Start the new sentence with „Future research are also needed to further…“.

-Line 263: Put „Figure 3.“ instead of „FIGURE3“.

-Line 265: Put „x“ instead of „*“.

REFERENCES

-Line 286: It should be Science.

-Line 288: It should be Sci. Agric. Sin.

-Line 292: Chin. J. Biol. Control

-Line 293: „Bradisia“ instead of „bradisia“.

-Line 294: Plant Protect.

-Line 296: I „gosypii“ final i is not in italic. Journal abbreviation should be Acta Ecol. Sin.

-Lines 304-305: J. Meteorol. Res.

-Line 307: Chin. J. Biol. Control

-Line 309: J. Zhejiang Univ. - Agric. Life Sci.

-Line 317: Plant Protect.

-Line 321: beetles’. Apostrophe is too small.

-Line 323: Sci. Agric. Sin.

-Line 325: Delete one dash line.

-Line 328: Put „China“ instead of „china“.

-Line 329: Chin. J. Appl. Entomol.

-Line 336: Sci. Agric. Sin.

-Line 345: Bull. Entomol. Res.

-Line 347: Acta Ecol. Sin.

-Line 351: Plant Protect.

-Line 353: Acta Prataculturae Sin.

-Line 358: Acta Entomol. Sin.

-Line 360: Acta Entomol. Sin.

-Line 362: Acta Agri. Boreali-occidentalis Sin.

-Line 366: Cell Stress Chaperones

Author Response

Response to Comments

Point 1: -There are still many missing spaces (e.g., lines 9, 72, 144, 156, 159, 162, 167, 172, 179, 248, 342, Y titles in Figure 1…

Response 1: They have been modified.

Point 2: -Lines 15-16: Chnange sentence into “These studies can contribute to deeper understanding how short-term heat exposure change CPB fitness.“

-Line 24: Put “…then those of control (CK) beetles“. Abbreviation CK appears for the first time in the text.

-Line 33: Put „comparing to endotherms“ at the end of sentence.

-Line 49: Do not start the sentence with the number.

- Lines 80-82: The sentence should not be in parenthesis. Put the separate sentence at the end of paragraph. Similarly the part in lines 85-93 should not be in parenthesis.

-Line 124: Correct“viposition“.

-Line 125: Is it SigmaPlot instead of SigmPlot?

-Line 127: Comma instead of full stop before “net“.

-Line 137: Replace“analyze“ with “we analyzed“.

-Line 141: Put p˂0.001.

-Line 145: Put “exposed to“ before “repeated“. Put“Short-term“ instead of “short-term“.

-Lines 15, 161, 162, 212, 215, : Delete “by“.

-Line 163: Sign for female is not needed.

-Line 22: Delete “by“, put “population“ before“increase“.

-Line 194: Delete “4. Discuss“ at the beginning of subtitle.

-Lines 204-205: Start with “It has been also found that the emergence rate….“

-Line 207: “Starting from 44℃ CPB survival rate…” should be new sentence.

-Line 208: Insert“exposure“ before „time“ and „in“ before „the first instar“. Replace“so“ with “pointing that“.

-Line 212: Replase “the same as“ with “in accordance with“.

-Line 213: Delete “per“.

-Line 216: Insert“showing“ before “that“.

-Line 218: “It shows that short-term….“ should be new sentence.

-Lines 227-229: Put “disturb other physiological and behavioral traits“ instead of „destroy…“. Cite some references at the end of the sentence.

-Line 231: Delete one “the“. Replace “raised“ with “reared“.

-Line 232: Part in parentheses can be deleted.

-Line 234: Change into: “After exposure of CPB eggs to short-term….“.

-Line 235: Change into: “After exposure of CPB adults to short-term….“.

-Lines 238-240: Change into: “Similar to our results repeated high-temperature exposure of…“.

-Lines 240-242: The sentence is not clear.

-Line 245: Put“temperatures“ instead of “temperature“. Put “and“ between 35 and 39. Put “higher“ instead of „above“.

-Line 255: Start the new sentence with“Future research are also needed to further…“.

-Line 263: Put “Figure 3.“ instead of „FIGURE3“.

-Line 265: Put “x“ instead of „*“.

Response 2: They have been modified.

Point 3: REFERENCES

-Line 286: It should be Science.

-Line 288: It should be Sci. Agric. Sin.

-Line 292: Chin. J. Biol. Control

-Line 293: “Bradisia“ instead of “bradisia“.

-Line 294: Plant Protect.

-Line 296: I “gosypii“ final i is not in italic. Journal abbreviation should be Acta Ecol. Sin.

-Lines 304-305: J. Meteorol. Res.

-Line 307: Chin. J. Biol. Control

-Line 309: J. Zhejiang Univ. - Agric. Life Sci.

-Line 317: Plant Protect.

-Line 321: beetles’. Apostrophe is too small.

-Line 323: Sci. Agric. Sin.

-Line 325: Delete one dash line.

-Line 328: Put “China“ instead of“china“.

-Line 329: Chin. J. Appl. Entomol.

-Line 336: Sci. Agric. Sin.

-Line 345: Bull. Entomol. Res.

-Line 347: Acta Ecol. Sin.

-Line 351: Plant Protect.

-Line 353: Acta Prataculturae Sin.

-Line 358: Acta Entomol. Sin.

-Line 360: Acta Entomol. Sin.

-Line 362: Acta Agri. Boreali-occidentalis Sin.

-Line 366: Cell Stress Chaperones

Response 3: They have been modified.

Point 4:

-Line 125: Which post hoc test did you use for non-parametric analyses?

-Lines 168, 191: LSD is not post hoc test for non-parametric analyses.

Response 4: The data of demographic parameters and oviposition traits for the CPBs that used Kruskal-Wallis test, and all pairwise multiple comparison was used to conduct a post-hoc test. The p value is the Holm-Bonferroni Correction value.
